# Prediction of final pathology depending on preoperative myometrial invasion and grade assessment in low-risk endometrial cancer patients: A Korean Gynecologic Oncology Group ancillary study

Dong-hoon Jang[1]☉, Hyun-Gyu Lee[1,2]☉, Banghyun Lee[3]*, Sokbom Kang[4], Jong-Hyeok Kim[5], Byoung-Gie Kim[6], Jae-Weon Kim[7], Moon-Hong Kim[8], Xiaojun Chen[9], Jae Hong No[10], Jong-Min Lee[11], Jae-Hoon Kim[12], Hidemich Watari[13], Seok Mo Kim[14], Sung Hoon Kim[15], Seok Ju Seong[16], Dae Hoon Jeong[17], Yun Hwan Kim[18]

1 Department of Electrical and Computer Engineering, Inha University, Incheon, Republic of Korea,
2 College of Medicine, Inha University, Incheon, Republic of Korea, 3 Department of Obstetrics and Gynecology, Inha University hospital, Inha University College of Medicine, Incheon, Republic of Korea, 4 Gynecologic Oncology Research Branch, Research Institute and Hospital, National Cancer Center, Goyang, Republic of Korea, 5 Department of Obstetrics and Gynecology, University of Ulsan College of Medicine, Asan Medical Center, Seoul, Republic of Korea, 6 Department of Obstetrics and Gynecology, Samsung Medical Center, Sungkyunkwan University School of Medicine, Seoul, Republic of Korea, 7 Department of Obstetrics and Gynecology, College of Medicine, Seoul National University, Seoul, Republic of Korea, 8 Department of Obstetrics and Gynecology, Korea Cancer Center Hospital, Korea Institute of Radiological and Medical Sciences, Seoul, Republic of Korea, 9 Department of Gynecology, Obstetrics and Gynecology Hospital of Fudan University, Shanghai, China, 10 Department of Obstetrics and Gynecology, Seoul National University Bundang Hospital, Seongnam, Republic of Korea, 11 Department of Obstetrics and Gynecology, College of Medicine, Kyung Hee University Hospital at Gangdong Kyung Hee University, Seoul, Republic of Korea, 12 Department of Obstetrics and Gynecology, Gangnam Severance Hospital, Yonsei University College of Medicine, Seoul, Republic of Korea, 13 Department of Gynecology, Graduate School of Medicine, Hokkaido University, Sapporo, Japan, 14 Department of Obstetrics and Gynecology, Chonnam National University Medical School, Gwangju, Republic of Korea, 15 Department of Obstetrics and Gynecology, Institute of Women's Life Science, Yonsei University College of Medicine, Seoul, Republic of Korea, 16 Department of Obstetrics and Gynecology, CHA Gangnam Medical Center, CHA University, Seoul, Republic of Korea, 17 Department of Obstetrics and Gynecology, Busan Paik Hospital, College of Medicine, Inje University, Busan, Republic of Korea, 18 Department of Obstetrics and Gynecology, Ewha Womans University Mokdong Hospital, Ewha Womans University College of Medicine, Seoul, Korea

☉ These authors contributed equally to this work.
* banghyun.lee@gmail.com

## Abstract

### Objectives

Fertility-sparing treatment (FST) might be considered an option for reproductive patients with low-risk endometrial cancer (EC). On the other hand, the matching rates between preoperative assessment and postoperative pathology in low-risk EC patients are not high enough. We aimed to predict the postoperative pathology depending on preoperative myometrial invasion (MI) and grade in low-risk EC patients to help extend the current criteria for FST.

**Data Availability Statement:** All relevant data are included in the manuscript and its Supporting

Information files. S2 Table lists the raw data. The code used for this study can be accessed at https://github.com/MAI00024/2024-endometrium.

**Funding:** The author(s) received no specific funding for this work.

**Competing interests:** The authors have declared that no competing interests exist.

## Methods/Materials

This ancillary study (KGOG 2015S) of Korean Gynecologic Oncology Group 2015, a prospective, multicenter study included patients with no MI or MI <1/2 on preoperative MRI and endometrioid adenocarcinoma and grade 1 or 2 on endometrial biopsy. Among the eligible patients, Groups 1–4 were defined with no MI and grade 1, no MI and grade 2, MI <1/2 and grade 1, and MI <1/2 and grade 2, respectively. New prediction models using machine learning were developed.

## Results

Among 251 eligible patients, Groups 1–4 included 106, 41, 74, and 30 patients, respectively. The new prediction models showed superior prediction values to those from conventional analysis. In the new prediction models, the best NPV, sensitivity, and AUC of preoperative each group to predict postoperative each group were as follows: 87.2%, 71.6%, and 0.732 (Group 1); 97.6%, 78.6%, and 0.656 (Group 2); 71.3%, 78.6% and 0.588 (Group 3); 91.8%, 64.9%, and 0.676% (Group 4).

## Conclusions

In low-risk EC patients, the prediction of postoperative pathology was ineffective, but the new prediction models provided a better prediction.

## Introduction

Continuous progestin-based therapy can be considered for selected patients with early-stage disease who wish to preserve their fertility despite the primary treatment of endometrial cancer (EC) being a hysterectomy. The criteria for considering fertility-sparing options in the NCCN Guidelines for EC include biopsy-proven grade 1 endometrioid adenocarcinoma, no myometrial invasion (MI) on magnetic resonance imaging (MRI) or transvaginal sonography, and an absence of suspicious or metastatic disease on imaging [1]. On the other hand, fertility-sparing treatment (FST) might be considered a valid option for reproductive-aged patients with grade 1 or 2 endometrioid adenocarcinoma and no MI or MI <1/2 because they are commonly considered a "low-risk" population [2–5]. In low-risk populations, according to various criteria, the risk of lymph node metastasis ranged between 1.7% and 2.9% [6,7].

Whether EC patients are candidates for FST is usually determined by MRI and dilatation and curettage [1]. In previous studies, however, the matching rates between the preoperative assessment and postoperative pathology were not high enough, causing concern about extending the candidates for FST [7–12]. MRI is considered a suitable tool for the preoperative staging of ECs. On the other hand, the MRI scan has shown a broad range of predictive values in assessing MI [8–11]. In relation to the presence or absence of MI, the accuracy, negative predictive value (NPV), and sensitivity were 54.8–86%, 35–87% (46–95% for grade 1), and 94%, respectively [8]. The matching rates between the preoperative MRI and postoperative pathology were 28.6–84.2% for no MI and 51.2–64.4% for MI <1/2 [8–11]. The matching rates of the grades between a preoperative endometrial biopsy and postoperative pathology were 74.8–94.4% for grade 1 and 43.8–58.8% for grade 2 [7,9,12].

If previous studies reported a relationship between preoperative assessment based on the MRI and biopsy results and the postoperative final pathology in patients with low-risk EC,

they were performed in retrospective and small studies [8–11]. Moreover, the incidence of EC has increased, particularly in young-reproductive aged patients who can be candidates for FST [13,14]. Therefore, those relationships need to be clarified in large-scale prospective studies. These results might help clarify whether the current criteria for FST in EC can be extended so that more young EC patients can receive FST.

This study examined the relationship between the preoperative assessment and postoperative final pathology depending on the presence or absence of MI and grades 1 or 2 in patients with low-risk EC on a preoperative assessment using the data from the Korean Gynecologic Oncology Group (KGOG) 2015, which was a prospective, multicenter observational study [6].

## Materials and methods

### 1. Study population

In KGOG 2015, between January 1, 2012, and December 31, 2014, 529 consecutive EC patients underwent a preoperative assessment based on MRI, an endometrial biopsy, and serum CA 125 testing, followed by surgical staging, including systemic pelvic and para-aortic lymphadenectomy [6]. In this prospective, multicenter cohort study, the participants were enrolled in 20 hospitals in three countries (Korea, Japan, and China) between January 2012 and December 2014. Approval from local institutional review boards was obtained for each center. Each participating center was a tertiary hospital that regularly provided surgical care for EC and had multidisciplinary teams that included specialized gynecologic oncologists, pathologists, and radiologists. The inclusion criteria were as follows: 1) EC, 2) no deep MI (MI <1/2 on MRI, 3) no enlarged lymph nodes on MRI, 4) no suspicious extrauterine spread, and 5) serum CA 125 < 35 U/mL. Patients with squamous cell carcinoma or carcinosarcoma on a preoperative biopsy, inadequate imaging study, no lymph node dissection, or sarcoma were excluded. The 2009 FIGO classification was used for the stage based on the final pathological findings.

This ancillary study of KGOG 2015 (KGOG 2015S) was performed in accordance with the Korean Bioethics and Safety Act and approved by the Institutional Review Board of Inha University Hospital (No. 2021-09-024) on September 27, 2021. Written informed consent was obtained from the participants. Data were accessed for research purposes on July 1, 2022.

The inclusion criteria were no MI or MI <1/2 on preoperative MRI and endometrioid adenocarcinoma and grades 1 or 2 on the endometrial biopsy; there were no exclusion criteria.

### 2. Study groups, variables and efficacy assessment

The eligible patients were assigned to four groups (Fig 1). Group 1 included patients with no MI on preoperative MRI and grade 1 on an endometrial biopsy. Group 2 included patients with no MI on preoperative MRI and grade 2 on an endometrial biopsy. Group 3 included patients with MI <1/2 on preoperative MRI and grade 1 on an endometrial biopsy. Group 4 included patients with MI <1/2 on preoperative MRI and grade 2 on an endometrial biopsy.

In Groups 1–4 and the total eligible patients, the following variables were evaluated: age, menopause, methods of preoperative endometrial biopsy, endometrial biopsy-proven grade, MI and tumor size (largest diameter) on MRI, preoperative serum CA125, methods of surgery, surgical stage, histologic diagnosis, grade, MI, tumor size, lymphovascular space invasion (LVSI), extrauterine involvement and metastasis to pelvic lymph node, matching rates of MI, grades and groups between the preoperative assessment and postoperative final pathology; NPV, positive predictive value (PPV), sensitivity, specificity, and area under the curve (AUC) of preoperative assessment to predict the final postoperative pathology by conventional analysis or using machine learning technology, including new models.

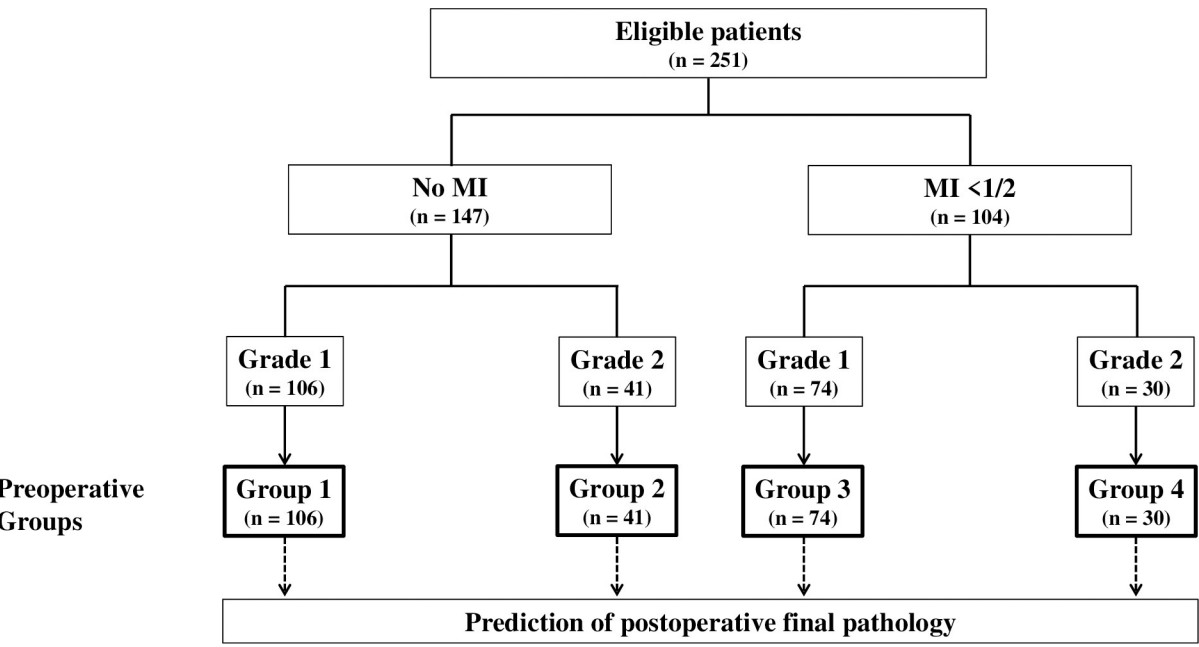

**Fig 1. Classification of the study groups according to the preoperative MI and grade assessment.**

## 3. Proposed methods

This study introduced two novel models to predict postoperative pathology depending on the preoperative MI and grade among low-risk EC patients. Their performance was compared with three conventional machine learning algorithms: logistic regression, extreme gradient boosting and support vector machines. Deep learning-based algorithms were not explored because of the limited training data.

**3.1. New Prediction Models (NPMs).** Two prediction models, NPM1 and NPM2, were proposed. Both were designed around the principal variable, the depth of MI. In NPM1, the depth of MI served as the principal variable. The model used iterative imputation techniques to address the discrepancies observed in MI depth diagnosis results. NPM2 removed the inaccurate MI depth information and used label smoothing to improve prediction accuracy.

*3.1.1. Data preprocessing.* Some data processing was performed before training the model. First, missing values were identified in the preoperative tumor size (largest diameter) (Table 1). All these missing values were imputed with mean postoperative tumor size values, which were stratified by categories of the depth of postoperative MI. The depth of postoperative MI contained three categories: no, $<1/2$, and $\geq 1/2$. The mean postoperative tumor size (largest diameter) for each category was 1.41 cm, 2.45 cm, and 2.95 cm, respectively.

*3.1.2. New prediction model 1 (NPM 1)*
### TRAINING SET

Fig 2A shows the proposed methodology and the overall workflow of the model training process. The imputation technique was used to calibrate the discrepancies between the preoperative and postoperative depth of MI, the key variable. The dataset had a class imbalance issue. This class imbalance was overcome by splitting the data into equal class proportions and organizing multiple sub-datasets. For example, in the case of Group 1, a total of three sub-datasets were generated if Group 1 and the three other groups (Groups 2, 3, and 4) had a ratio of 1:3.

**Table 1. Characteristics of the study population according to the study groups.**

| | Total n = 251 (100%) | Group 1 n = 106 (42.2%) | Group 2 n = 41 (16.3%) | Group 3 n = 74 (29.5%) | Group 4 n = 30 (12.0%) |
|---|---|---|---|---|---|
| Age, mean ± SD, y | 52.8 ± 9.6 | 51.3 ± 8.9 | 52.1 ± 8.6 | 53.4 ± 9.6 | 52.8 ± 8.4 |
| Menopause, n (%) | 152 (60.6) | 58 (54.7) | 29 (70.7) | 44 (59.5) | 21 (70.0) |
| **Preoperative endometrial biopsy** | | | | | |
| Methods, n (%) | | | | | |
| Dilatation and curettage | 228 (90.8) | 95 (89.6) | 38 (92.7) | 70 (94.6) | 25 (83.3) |
| Hysteroscopy | 6 (2.4) | 4 (3.8) | 0 (0) | 1 (1.4) | 1 (3.3) |
| Pipelle biopsy | 17 (6.8) | 7 (6.6) | 3 (7.3) | 3 (4.1) | 4 (13.3) |
| Grade, n (%) | | | | | |
| 1 | 180 (71.7) | 106 (100) | 0 (0) | 74 (100) | 0 (0) |
| 2 | 71 (28.3) | 0 (0) | 41 (100) | 0 (0) | 30 (100) |
| **Preoperative MRI** | | | | | |
| MI, n (%) | | | | | |
| No | 147 (58.6) | 106 (100) | 41 (100) | 0 (0) | 0 (0) |
| <1/2 | 104 (41.4) | 0 (0) | 0 (0) | 74 (100) | 30 (100) |
| Tumor size (largest diameter), n (%) | | | | | |
| <2cm | 78 (31.1) | 38 (35.9) | 14 (34.2) | 17 (23) | 9 (30) |
| ≥2cm | 84 (33.5) | 25 (23.6) | 8 (19.5) | 38 (51.4) | 13 (43.3) |
| Missing value | 89 (35.5) | 43 (40.6) | 19 (46.3) | 19 (25.7) | 8 (26.7) |
| Preoperative serum CA125 (IU/ml) | 19.2 ± 10.1 | 21.2 ± 9.8 | 17.7 ± 9.3 | 20.4 ± 11.6 | 25.7 ± 38.3 |
| **Methods of surgery, n (%)** | | | | | |
| Laparoscopy or robotic | 216 (86.1) | 94 (88.7) | 35 (85.4) | 61 (82.4) | 26 (86.7) |
| Laparotomy | 35 (13.9) | 12 (11.3) | 6 (14.6) | 13 (17.6) | 4 (13.3) |
| **Postoperative pathologic findings** | | | | | |
| Surgical stage[a], n (%) | | | | | |
| IA | 227 (90.4) | 100 (94.3) | 36 (87.8) | 68 (91.9) | 23 (76.7) |
| IB | 12 (4.8) | 3 (2.8) | 0 (0) | 5 (6.8) | 4 (13.3) |
| II | 6 (2.4) | 0 (0) | 2 (4.9) | 1 (1.4) | 3 (10) |
| IIIA | 0 (0) | 0 (0) | 0 (0) | 0 (0) | 0 (0) |
| IIIB | 0 (0) | 0 (0) | 0 (0) | 0 (0) | 0 (0) |
| IIIC | 6 (2.4) | 3 (2.8) | 3 (7.3) | 0 (0) | 0 (0) |
| IV | 0 (0) | 0 (0) | 0 (0) | 0 (0) | 0 (0) |
| Histologic diagnosis, n (%) | | | | | |
| Endometrioid | 248 (98.8) | 105 (99.1) | 40 (97.6) | 73 (98.6) | 30 (100) |
| Non-endometrioid | 3 (1.2) | 1 (0.9) (Mixed with endometrioid) | 1 (2.4) (Squamous or adenosquamous) | 1 (1.4) (Mixed with endometrioid) | 0 (0) |
| Grade, n (%) | | | | | |
| 1 | 177 (70.5) | 88 (83.0) | 16 (39.0) | 64 (86.5) | 9 (30) |
| 2 | 63 (25.1) | 13 (12.3) | 20 (48.8) | 10 (13.5) | 20 (66.7) |
| 3 | 5 (2.0) | 2 (1.9) | 2 (4.9) | 0 (0) | 1 (3.3) |
| Inadequate for interpretation | 6 (2.4) | 3 (2.8) | 3 (7.3) | 0 (0) | 0 (0) |
| MI, n (%) | | | | | |
| No | 91 (36.3) | 57 (53.8) | 14 (34.1) | 14 (18.9) | 6 (20) |
| <1/2 | 147 (58.6) | 46 (43.4) | 26 (63.4) | 55 (74.3) | 20 (66.7) |
| ≥1/2 | 13 (5.2) | 3 (2.8) | 1 (2.4) | 5 (6.8) | 4 (13.3) |
| Tumor size (largest diameter), n (%) | | | | | |
| <2cm | 127 (50.6) | 61 (57.6) | 22 (53.7) | 20 (27.0) | 14 (46.7) |

*(Continued)*

**Table 1.** (Continued)

| | Total n = 251 (100%) | Group 1 n = 106 (42.2%) | Group 2 n = 41 (16.3%) | Group 3 n = 74 (29.5%) | Group 4 n = 30 (12.0%) |
|---|---|---|---|---|---|
| ≥2cm | 124 (49.4) | 45 (42.5) | 19 (46.3) | 54 (73.0) | 16 (53.3) |
| **LVSI, n (%)** | | | | | |
| **No** | 229 (91.2) | 103 (97.2) | 35 (85.4) | 67 (90.5) | 24 (80) |
| **Yes** | 21 (8.8) | 3 (2.8) | 6 (14.6) | 7 (9.5) | 6 (20) |
| **Extrauterine involvement, n (%)** | | | | | |
| **No** | 245 (97.6) | 0 (0) | 39 (95.1) | 73 (98.6) | 27 (90) |
| **Yes** | 6 (2.4) (cervix) | 0 (0) | 2 (4.9) | 1 (1.4) | 3 (10) |
| **Metastasis to pelvic lymph node, n (%)** | | | | | |
| **No** | 245 (97.6) | 103 (97.2) | 38 (92.7) | 74 (100) | 30 (100) |
| **Yes** | 6 (2.4) | 3 (2.8) | 3 (7.3) | 0 (0) | 0 (0) |

[a] The 2009 FIGO classification was used for the surgical stage.

LVSI, lymphovascular space invasion; MI, myometrial invasion; MRI, magnetic resonance imaging.

For each generated sub-dataset, the K-Fold technique (K = 5) was used to partition the data. Each partitioned sub-dataset was divided into training and validation datasets and multiple weak classifiers were trained using those datasets. Among the trained models, the best models were selected according to the proposed evaluation criteria (Eq (1)) using the validation data of each sub-dataset, and the ensemble model was developed by integrating those.

## Missing Value Imputation Techniques

The matching rates for the primary variables, depth of MI and grade, between the preoperative and postoperative stages were low. Such discrepancies in these primary metrics could adversely affect the performance of the predictive model. This concern was addressed by applying the imputation techniques of the present study to rectify the inconsistencies between preoperative and postoperative assessments of MI depth [15]. This study used the 'Iterative Imputation' based on Linear Regression, a derivative of the Multivariate Imputation by Chained Equations (MICE) technique [16]. MICE is a statistical technique that uses multivariate regression equations to calibrate the values, using the relationship among variables to calibrate value.

The calibration of the MI depth was addressed by training the proposed imputation model on a set comprising seven variables, including the original MI depth values. Within the training set, this imputation model was applied specifically to rectify the inconsistencies between pre- and postoperative measurements of the MI depth. The MI depth values were adjusted using the trained imputation model throughout the training process, while the corresponding grade values remained unaltered. In the experiment, higher matching rates between the preoperative and postoperative key variables, such as the depth of MI, have been shown to improve the model performance (S1 Fig). Therefore, enhancing the matching rates for these variables is crucial. When comparing the matching rates of the depth of MI and grade, the average matching rates for the depth of MI were 58.2%, with 'no MI' at 48.3% and 'MI <1/2%' at 72.1%. On the other hand, for the grade, 'grade 1' and 'grade 2' had a matching rate of 84.4% and 56.3%, respectively; the average matching rate was 76.5%. The depth of MI was treated as missing because the matching rates of the depth of MI were lower than the grade, and the values were replaced using the trained missing value imputation model (Fig 2A).

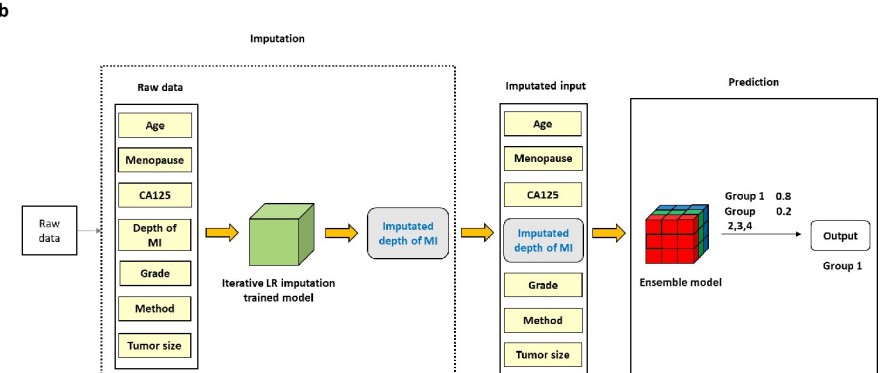

**Fig 2. NPM1**. a. Study overview in the training set. b. Study overview in the test set. The experiment was repeated in the following classifications: in Group 1 and Group 2, 3, and 4; in Group 2 and Groups 1, 3, and 4; in Group 3 and Groups 1, 2, and 4; in Group 4 and Groups 1, 2, and 3.

## Constructing Data to Address the Class Imbalance Problem

Class imbalance significantly impacts the performance of classification tasks. Machine learning algorithms are often biased towards the majority class to maximize the overall accuracy, resulting in a poor prediction for minority classes [17]. Cross-validation techniques, such as K-Fold are commonly used to measure the generalization performance of a model [18]. On the other hand, Stratified K-Fold Cross-Validation (SKCV) was particularly well-suited for dealing with

unbalanced class distributions. Unlike the standard K-Fold, SKCV maintains the class proportions in each training and validation fold, providing a more balanced representation [19,20].

A two-tiered approach involving SKCV and K-Fold techniques was used to tackle the class imbalance issue in a dataset with one specific group and three others in this study. First, the entire dataset was split into training and test datasets using the Stratified K-Fold technique (K = 5). Within the training dataset, the data was divided considering relative class proportions, and sub-datasets with the same class proportion were generated ($D_1$, $D_2$, and $D_3$). The K-Fold technique (K = 5) was used to partition the data for each generated sub-dataset, and the model was evaluated using the validation data from the partitioned sub-dataset. The experiment was replicated across various classifications: between Group 1 and the other groups (Groups 2, 3, and 4), between Group 2 and the other groups (Groups 1, 3, and 4), between Group 3 and the other groups (Groups 1, 2, and 4), and between Group 4 and the other groups (Groups 1, 2, and 3) (Fig 2A).

## Constructing Ensemble Models through Weak Classifier Training

Ensemble selection techniques are valuable tools for constructing ensembles from a large collection of classifiers, achieving superior performance [21]. These techniques work by carefully choosing models from an abundant set of classifiers to form a more effective ensemble [22]. Combining various classifiers selectively can often outperform any single model.

In this study, the class imbalance challenge was tackled using an ensemble approach. In particular, this study generated tailored, class-balanced subsets of the data on which multiple weak classifiers were trained. Each of these classifiers was evaluated on separate validation sets. The final ensemble model was constructed by combining selected models (SM), guided by the following equation:

$$SM = \underset{M_{ij}}{\mathrm{argmax}}((TP + TN) - (FP + FN)) \tag{1}$$

*TP*: True Positives, *TN*: True Negatives, *FP*: False Positives, and *FN*: False Negatives.

The experiments were conducted to assess the performance in different scenarios: between Group 1 and the remaining groups (Groups 2, 3, and 4), between Group 2 and the remaining groups (Groups 1, 3, and 4), between Group 3 and the remaining groups (Groups 1, 2, and 4), and finally between Group 4 and the remaining groups (Groups 1, 2, and 3) (Fig 2A).

## TEST SET

In the testing phase, the imputation model trained during the training stage was applied to calibrate the depth of MI. The calibrated depth of MI and six other variables were used as input for the ensemble model, also trained earlier, to make predictions (Fig 2B).

*3.1.3. New prediction model 2 (NPM 2).* Unlike NPM1, NPM2 did not use the depth of MI as a variable in the imputation process (Fig 3). All other experimental conditions were the same for both models (Figs 2 and 3). In NPM1, the depth of MI was generally calibrated to values close to 0 or 1 (S2 Fig). This calibration effect was attributed to the use of the depth of MI as a variable, which has the highest correlation with the imputed depth of MI. In contrast, in NPM2, the imputed values were smoothly compensated between 0 and 1 (S2 Fig).

NPM2 outperformed NPM1 for several reasons. First, the depth of MI was categorized as no MI, MI <1/2, and MI ≥1/2 in the dataset. On the other hand, the depth of MI of each patient differed among patients in the same category. Therefore, rather than categorizing patients into these three categories, a continuous representation of the depth of MI for each patient would reflect the variability of the stage more accurately. Second, labeling smoothing,

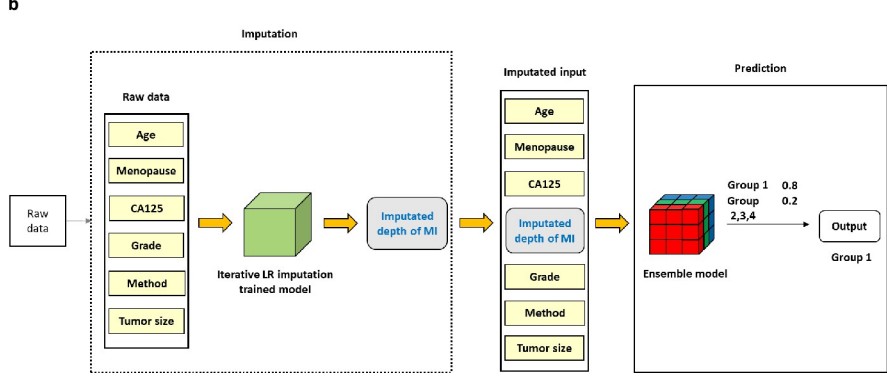

**Fig 3. NPM2.** a. Study overview in the training set. b. Study overview in the test set. The experiment was repeated in the following classifications: in Group 1 and Group 2, 3, and 4; in Group 2 and Groups 1, 3, and 4; in Group 3 and Groups 1, 2, and 4; in Group 4 and Groups 1, 2, and 3. NPM 2, New Prediction Model 2; MI, myometrial invasion. The difference compared to NPM1 is that NPM2 did not use the depth of MI as a variable in the imputation process.

which converts hard labels to soft labels, was labeled [23]. This approach prevents overfitting by reducing the confidence level for the actual class and assigning minor probabilities to other classes, improving the generalization performance of the model [24].

The depth of MI was one of the essential variables in determining Groups 1–4. Better matching rates between the preoperative and postoperative depth of MI led to improved model performance (S1 Fig). The Mean Squared Error (MSE) metric was used to compare the

calibrated depth of MI with its postoperative values to determine if NPM2 improved the matching rates compared to NPM1. The equation for MSE is as follows:

$$\text{MSE} = \frac{1}{n}\sum\nolimits_{i=0}^{n}\left(Y_i - \hat{Y}_i\right)^2 \tag{2}$$

$Y_i$: postoperative depth of MI, $\hat{Y}_i$: calibrated depth of MI.

The MSE values were 0.47 for NPM1 and 0.31 for NPM2, suggesting that NPM2 produced calibrated depths of MI more closely aligned with their postoperative values. This result confirmed that NPM2 was a more effective way of predicting the groups than NPM1.

**3.2. Comparison algorithms.** *3.2.1. Logistic regression.* Logistic Regression is a classification algorithm based on linear regression, used primarily for binary classification problems [25]. This method is suitable when the dependent variable is categorical data. This algorithm calculates the sum of the product of independent variables and weights, similar to linear regression. The logistic function (sigmoid function) converts the probability to a value between 0 and 1. Based on this probability, the threshold was set to predict the class. Logistic regression is a statistical modeling technique that identifies causal relationships between the variables based on the magnitude of the coefficients and their associated odds ratios [26].

*3.2.2. Extreme gradient boosting.* Extreme Gradient Boosting (XGBoost) is an ensemble learning algorithm derived from the foundational Gradient Boosting framework. The algorithm showed superior performance in both classification and regression challenges [27]. This method involves sequentially training weak learners, typically decision trees, to optimize weights iteratively and reduce the errors of their predecessors. XGBoost executes this procedure with high efficiency and speed and incorporates regularization terms to mitigate the risk of overfitting, enhancing model performance.

*3.2.3 Support vector machine.* A Support Vector Machine (SVM) is a supervised learning algorithm for classification tasks [28]. The algorithm seeks to maximize the margin, defined as the distance between the nearest data points (known as support vectors) from different classes to the decision boundary, typically a hyperplane. SVMs use kernel methods to transform the original data into a higher-dimensional feature space and handle nonlinear classification problems, wherein an optimal separating hyperplane is identified. SVMs often achieve commendable generalization performance using this margin maximization technique [29].

**3.3. Statistical analyses.** Descriptive analyses were performed to evaluate the matching, upstaging, downstaging, upgrading, and downgrading rates between preoperative assessment and postoperative final pathology. Moreover, the agreement levels between the preoperative and postoperative pathological findings were analyzed using Kappa statistics. Kappa values (k) were interpreted as follows: poor for $k \le 0.2$; fair for $0.2 < k \le 0.4$; moderate for $0.4 < k \le 0.6$; substantial for $0.6 < k \le 0.8$; good for $> 0.8$ [30]. The NPV, PPV, sensitivity, specificity, and AUC of preoperative assessment were calculated to predict postoperative final pathology. The analyses used Python 3.7.13 (Python Software Foundation, http://www.python.org). The 95% confidence intervals for Kappa and AUC were calculated using the bootstrap method. The bootstrap method was implemented directly using the resample function from the scikit-learn package (https://scikit-learn.org, version 1.0.2).

# Results

Two hundred and fifty-one eligible patients were selected from KGOG 2015 dataset, and 106 (42.2%), 41 (16.3%), 74 (29.5%), and 30 (12.0%) were assigned to Groups 1, 2, 3, and 4, respectively (Fig 1).

## 1. Characteristics of the study population according to study groups

Table 1 lists the baseline characteristics of the study subjects. On the endometrial biopsy, 180 (71.7%) patients had grade 1, and 71 (28.3%) had grade 2. On preoperative MRI, 147 (58.6%), 104 (41.4%), and 84 (33.5%) patients had no MI, MI <1/2, and tumor size ≥2cm, respectively. In the postoperative pathologic findings, 227 (90.4%), patients had stage IA, 12 (4.8%) stage IB, six (2.4%) stage II, six (2.4%) stage IIIC, 248 (98.8%) endometrioid type, three (1.2%) non-endometrioid type, 177 (70.5%) grade 1, 63 (25.1%) grade 2, five (2.0%) grade 3, 91 (36.3%) no MI, 147 (58.6%) MI <1/2, 13 (5.2%) MI ≥1/2, 124 (49.4%) tumor size ≥2cm, 21 (8.8%) LVSI, six (2.4%) cervical involvements, and six (2.4%) pelvic lymph node involvements.

## 2. Relationship between the preoperative assessment and postoperative final pathology according to the study groups

The matching rates between the preoperative assessment and postoperative final pathology were 43.4%, 14.6%, 60.8%, and 43.3% in Groups 1, 2, 3, and 4, respectively. According to the Kappa statistics, the agreements were poor to fair as follows: k = 0.304 (95% CI, 0.187–0.419) for Group 1; k = 0.145 (95% CI, 0.06–0.295) for Group 2; k = 0.281 (95% CI, 0.158–0.401) for Group 3; and k = 0.292 (95% CI, 0.128–0.453) for Group 4. In Group 1, 56.6% of patients were upstaged on postoperative pathology (5.7% in postoperative Group 2, 35.8% in postoperative Group 3, 4.7% in postoperative Group 4, 10.4% in higher stages), 75.6% in Group 2 (26.8% in postoperative Group 3, 24.4% in postoperative Group 4, and 24.4% at the higher stages), 20.2% in Group 3 (12.1% in postoperative Group 4, 8.1% at the higher stages), and 26.6% in Group 4 (26.6% at the higher stages). In Group 2, 9.7% of patients were downstaged on postoperative pathology (9.7% in postoperative Group 1), 18.9% in Group 3 (18.9% in postoperative Group 1), and 30.0% in Group 4 (10.0% in postoperative Group 1, 6.6% in postoperative Group 2, and 13.3% in postoperative Group 3) (Table 2). Higher stages included 4.4% of stage IA and grade 3, 4.8% of stage IB, 2.4% of stage II, and 2.4% of stage IIIC (S1 Table).

The matching rates of MI between the preoperative assessment and postoperative final pathology were 48.3% in no MI (53.8% in Group 1 and 34.1% in Group 2) and 72.1% in MI <1/2 (74.3% in Group 3 and 66.7% in Group 4). In Group 1, 46.2% of patients were upstaged on the postoperative pathology, 65.8% in Group 2, 6.8% in Group 3, and 13.3% in Group 4. Approximately 18.9% of patients in Group 3 and 20.0% in Group 4 were downstaged according to the postoperative pathology (Table 2).

The matching rates between the preoperative assessment and postoperative final pathology were 84.4% in grade 1 (83.0% in Group 1 and 48.8% in Group 2) and 56.3% in grade 2 (86.5% in Group 3 and 66.7% in Group 4). In Group 1, 14.2% of patients were upgraded on the postoperative pathology, 4.9% in Group 2, 13.5% in Group 3, and 3.3% in Group 4. Approximately 39.0% of patients in Group 2 and 30.0% in Group 4 were downgraded according to the postoperative pathology (Table 2).

## 3. Prediction of postoperative final pathology according to study groups

Table 3 lists the performance of preoperative assessment in predicting postoperative final pathology. The NPV, sensitivity, and AUC of preoperative each group to predict the postoperative each group were the highest in mainly NPMs as follows: 87.2% (NPM 2), 71.6% (NPM 1), and 0.732 (NPM 2) for Group 1; 97.6% (NPM 1), 78.6% (NPM 1), and 0.656 (NPM 2) for Group 2; 71.3% (NPM 2), 78.6% (NPM 1), and 0.635 (conventional analysis) for Group 3; and 91.8% (NPM 2), 64.9% (NPM 1), and 0.676% (NPM 2) for Group 4. PPV was the highest as follows: 61.1% for Group 1 (Logistic Regression), 14.6% and 60.8% in Groups 2 and 3

**Table 2. Relationship between the preoperative assessment and the postoperative final pathology according to the study groups.**

| | Preoperative classification | | | | | | | | | | | |
| | Total n = 251 (100%) | | | | Group 1 n = 106 (42.2%) | | Group 2 n = 41 (16.3%) | | Group 3 n = 74 (29.5%) | | Group 4 n = 30 (12.0%) | |
| | MI | | Grade | | MI | Grade | MI | Grade | MI | Grade | MI | Grade |
| | No n = 147 (58.6) | <1/2 n = 104 (41.4) | 1 n = 180 (71.1%) | 2 n = 71 (28.3%) | No | 1 | No | 2 | <1/2 | 1 | <1/2 | 2 |
| **Postoperative final pathology** | | | | | | | | | | | | |
| **MI, n (%)** | | | | | | | | | | | | |
| No | **71 (48.3)** | 20 (19.2) | | | **57 (53.8)** | | **14 (34.1)** | | 14 (18.9) | | 6 (20) | |
| <1/2 | 72 (49) | **75 (72.1)** | | | 46 (43.4) | | 26 (63.4) | | **55 (74.3)** | | **20 (66.7)** | |
| ≥1/2 | 4 (2.7) | 9 (8.7) | | | 3 (2.8) | | 1 (2.4) | | 5 (6.8) | | 4 (13.3) | |
| *Upstaged patients, n (%)* | *74 (51.7)* | *9 (8.7)* | | | *49 (46.2)* | | *27 (65.8)* | | *5 (6.8)* | | *4 (13.3)* | |
| *Downstaged patients, n (%)* | *0 (0.0)* | *20 (19.2)* | | | *0 (0.0)* | | *0 (0.0)* | | *14 (18.9)* | | *6 (20)* | |
| **Grade, n (%)** | | | | | | | | | | | | |
| 1 | | | **152 (84.4)** | 25 (35.2) | | **88 (83.0)** | | 16 (39.0) | | **64 (86.5)** | | 9 (30) |
| 2 | | | 23 (12.8) | **40 (56.3)** | | 13 (12.3) | | **20 (48.8)** | | 10 (13.5) | | **20 (66.7)** |
| 3 | | | 2 (1.9) | 3 (8.2) | | 2 (1.9) | | 2 (4.9) | | 0 (0.0) | | 1 (3.3) |
| **Inadequate for interpretation** | | | 3 (2.8) | 3 (7.3) | | 3 (2.8) | | 3 (7.3) | | 0 (0) | | 0 (0) |
| *Upgraded patients, n (%)* | | | *25 (14.7)* | *3 (8.2)* | | *15 (14.2)* | | *2 (4.9)* | | *10 (13.5)* | | *1 (3.3)* |
| *Downgraded patients, n (%)* | | | *0 (0.0)* | *25 (35.2)* | | *0 (0.0)* | | *16 (39.0)* | | *0 (0.0)* | | *9 (30)* |
| **Postoperative classification** | | | | | | | | | | | | |
| **Group 1, n (%)** | 67 (26.7) | | | | **46 (43.4)** | | 4 (9.7) | | 14 (18.9) | | 3 (10.0) | |
| **Group 2, n (%)** | 14 (5.6) | | | | 6 (5.7) | | **6 (14.6)** | | 0 (0.0) | | 2 (6.6) | |
| **Group 3, n (%)** | 98 (39.0) | | | | 38 (35.8) | | 11 (26.8) | | **45 (60.8)** | | 4 (13.3) | |
| **Group 4, n (%)** | 37 (14.7) | | | | 5 (4.7) | | 10 (24.4) | | 9 (12.1) | | **13 (43.3)** | |
| **Higher stages, n (%)[a]** | 35 (13.9) | | | | 11 (10.4) | | 10 (24.4) | | 6 (8.1) | | 8 (26.6) | |
| *Upstaged patients, n (%)* | *114 (45.9)* | | | | *60 (56.6)* | | *31 (75.6)* | | *15 (20.2)* | | *8 (26.6)* | |
| *Downstaged patients, n (%)* | *27 (10.8)* | | | | *0 (0.0)* | | *4 (9.7)* | | *14 (18.9)* | | *9 (30)* | |

[a] Higher stages: higher stages than postoperative Groups 1–4. Higher stages included stage 1A and grade 3, stage IB, stage II, and stage IIIC.

MI, myometrial invasion.

(conventional analysis), and 46.7% in Group 4 (SVM). The specificity was the highest in Logistic Regression as follows: 92.4% for Group 1, 100% in Group 2, 81.7% in Group 3, and 98.1% in Group 4.

## Discussion

In this ancillary study (KGOG 2015S) of KGOG 2015, a prospective, multicenter study, the low-risk EC patients on preoperative assessment were assigned into Groups 1–4 depending on

**Table 3. Prediction of the postoperative final pathology according to the study groups.**

| | Conventional analysis | Machine Learning | | | | |
|---|---|---|---|---|---|---|
| | | Logistic Regression | XGBoost | SVM | NPM1 | NPM2 |
| **Prediction of postoperative Group 1 by preoperative Group 1** | | | | | | |
| NPV | 85.5 | 79.1 | 78.6 | 78.4 | 85.9 | **87.2** |
| PPV | 43.4 | **61.1** | 33.3 | 48.9 | 41.4 | 52.9 |
| Sensitivity | 68.7 | 32.8 | 55.2 | 34.3 | **71.6** | 68.7 |
| Specificity | 67.4 | **92.4** | 59.8 | 87.0 | 63.0 | 77.7 |
| AUC (95% CI) | 0.681 (0.613–0.746) | 0.626 (0.566–0.686) | 0.575 (0.506–0.646) | 0.606 (0.544–0.668) | 0.674 (0.608–0.737) | **0.732 (0.666–0.793)** |
| **Prediction of postoperative Group 2 by preoperative Group 2** | | | | | | |
| NPV | 96.2 | 94.4 | 94.2 | 94.1 | **97.6** | 96.5 |
| PPV | **14.6** | N/A | 0.0 | 0.0 | 8.8 | 13.5 |
| Sensitivity | 42.9 | 0.0 | 0.0 | 0.0 | **78.6** | 50.0 |
| Specificity | 85.2 | **100** | 95.4 | 94.1 | 51.9 | 81.0 |
| AUC (95% CI) | 0.640 (0.506–0.779) | 0.500 (0.500–0.500) | 0.477 (0.462–0.489) | 0.470 (0.454–0.485) | 0.652 (0.522–0.759) | **0.656 (0.519–0.794)** |
| **Prediction of postoperative Group 3 by preoperative Group 3** | | | | | | |
| NPV | 70.1 | 68.3 | 71.2 | 66.9 | 70.8 | **71.3** |
| PPV | **60.8** | 58.8 | 48.1 | 52.6 | 43.0 | 46.0 |
| Sensitivity | 45.9 | 40.8 | 65.3 | 40.8 | **78.6** | 70.4 |
| Specificity | 81.0 | **81.7** | 54.9 | 76.5 | 33.3 | 47.1 |
| AUC (95% CI) | **0.635 (0.576–0.694)** | 0.613 (0.555–0.671) | 0.601 (0.539–0.662) | 0.586 (0.527–0.647) | 0.559 (0.504–0.615) | 0.588 (0.528–0.647) |
| **Prediction of postoperative Group 4 by preoperative Group 4** | | | | | | |
| NPV | 89.1 | 86.1 | 87.4 | 87.3 | 91.2 | **91.8** |
| PPV | 43.3 | 42.9 | 24.4 | **46.7** | 23.3 | 28.4 |
| Sensitivity | 35.1 | 8.1 | 29.7 | 18.9 | **64.9** | 62.2 |
| Specificity | 92.1 | **98.1** | 84.1 | 96.3 | 63.1 | 72.9 |
| AUC (95% CI) | 0.636 (0.557–0.718) | 0.531 (0.491–0.581) | 0.569 (0.493–0.650) | 0.576 (0.516–0.644) | 0.640 (0.554–0.722) | **0.676 (0.588–0.761)** |

AUC, area under the curve; CI, confidence interval; NPM 1, New Prediction Model 1; NPM 2, New Prediction Model 2; NPV, negative predictive value; PPV, positive predictive value; SVM, Support Vector Machine; XGBoost, Extreme Gradient Boosting.

the presence or absence of MI and grade 1 or 2. The matching rates between the preoperative assessment and postoperative final pathology were not high in patients included in Groups 1–4, no MI, MI <1/2, grade 1, and grade 2. Moreover, the agreements in Groups 1–4 were poor to fair. The prediction rates for postoperative pathology were not high enough in Groups 1–4. Compared with conventional analysis, the NPV, sensitivity, and AUC of each group preoperatively to predict each group postoperatively were the highest in NPMs, and the specificity was the highest in Logistic Regression.

This study examined the possibility of extending the current criteria for FST. Therefore, low-risk EC patients on preoperative assessment were classified into Groups 1–4. On the other hand, among the 251 eligible patients, most patients (42.2%) were included in Group 1 (no MI and grade 1), which was eligible for the current criteria for FST. The other Groups included

relatively fewer patients (16.3% for Group 2 (no MI and grade 2), 29.5% for Group 3 (MI <1/2 and grade 1), and 12.0% for Group 4 (MI <1/2 and grade 2)). These small sample sizes might negatively influence the predicted postoperative pathology in each group.

In previous retrospective and small studies performed on low-risk EC patients, the matching rates of MI and grade between preoperative assessment (MRI and endometrial biopsy) and the postoperative final pathology were unsatisfactory. They had a broad range (28.6–84.2% for no MI, 51.2–64.4% for MI <1/2, 94.4% for grade 1, and 43.8% for grade 2) [8–11]. This study showed similar matching rates to previous studies (48.3% in no MI, 72.1% in MI <1/2, 84.4% in grade 1, and 56.3% in grade 2). Moreover, this study reported low matching rates and poor to fair agreement in Groups 1–4. The matching rates were the lowest (14.6%) in Group 2 and the highest (60.8%) in Group 3, showing the same results when analyzing the matching rates of MI and grade in Groups 1–4. In Groups 1–4, the agreement was the lowest in Group 2 and similar in the other groups. In addition, in Group 2, 75.6% of patients were upstaged on postoperative pathology, and 9.7% were downstaged. These negative findings might be attributed to the smallest number of samples (n = 41) in Group 2. On the other hand, these findings have importance because Group 2 (no MI and grade 2) is the main candidate for criteria extension as the next step of Group 1 (no MI and grade 1), which is currently eligible for FST. Unfortunately, the patients in Groups 1–4 were in the high-frequency upstaged or downstaged group on the postoperative pathology. In particular, in Group 1, 56.6% of patients were upstaged, and 9.7%, 18.9%, and 10.0% of patients in Groups 2–4 were downstaged to postoperative Group 1, reducing the reliability of the current criteria for FST. On the postoperative pathology, the rates of patients in higher stages than postoperative Groups 1–4 were high and similar in Groups 1 and 3, and in Groups 2 and 4 depending on the grade (10.4% in Group 1, 24.4% in Group 2, 8.1% in Group 3, and 26.6% in Group 4). These findings also suggest that extending the current criteria for FST to Group 2 with grade 2 might be difficult.

The prediction rates of postoperative pathology were not high enough in Groups 1–4. The best NPV, sensitivity, and AUC of preoperative each group to predict the postoperative each group were as follows: 87.2%, 71.6%, and 0.732 for Group 1; 97.6%, 78.6%, and 0.656 for Group 2; 71.3%, 78.6% and 0.635 for Group 3; 91.8%, 64.9%, and 0.676% for Group 4. These findings induce concern in relation to the reliability of current criteria for FST and also suggest that it might be challenging to extend the current criteria for FST. Machine learning methods showed better prediction values than those from conventional analysis. In Groups 1, 2, and 4, the AUCs of each preoperative group were the highest in NPM2 but highest in Group 3 in conventional analysis. In Groups 1, 3, and 4, the NPVs were the highest in NPM2 and highest in NPM1 in Group 2. In Groups 1–4, the sensitivities were the highest in NPM1, and the specificities were the highest in Logistic Regression. In Groups 2 and 3, PPVs were the highest in conventional analysis and highest in Logistic Regression and SVM in Groups 1 and 4, respectively. The findings in which NPMs had the highest NPVs, sensitivities, and AUCs in Groups 1–4 showed that NPMs might be the best tools for predicting the postoperative outcome in each group depending on the presence or absence of MI and grades 1 or 2 among the low-risk EC patients on preoperative assessment.

The prediction abilities of NPM1 and NPM2 for the postoperative groups in Groups 1–4 were difficult to compare, even though NPM2 was considered more effective than NPM1. Moreover, the best prediction values were obtained when both were used. Therefore, NPM1 and NPM2 should be used together as prediction models. These NPMs showed better prediction abilities than the other Machine Learning methods.

The relevance of the present study stems from the prediction of postoperative pathology depending on the presence or absence of MI and grades 1 or 2 in low-risk EC patients on the preoperative assessment using a new machine learning model to help extend the current

criteria for FST. To the best of the authors' knowledge, this is the first study to develop prediction models for the postoperative pathology depending on the presence or absence of MI and grade 1 or 2 in low-risk EC patients on a preoperative assessment. Nevertheless, this study has some limitations. First, in KGOG 2015, a central review for MRI and pathology was not performed. Therefore, the MRI and pathology findings were not reviewed centrally in this ancillary study. Second, this study classified 251 eligible patients into Groups 1–4 with a small sample size. Hence, these small sample sizes could reduce the predictive abilities of NPMs. Third, this study included premenopausal (39.4%) and postmenopausal (60.6%) women despite the objective of this study being to extend the current criteria for FST in EC. On the other hand, EC commonly affects postmenopausal women, and the incidence of EC in premenopausal women is low (15–25%) [14]. Therefore, it is difficult to conduct the study with only premenopausal women. Moreover, there is no evidence that in EC, the pathology findings (endometrioid type or grade) or MRI findings differ between premenopausal women and postmenopausal women or are diagnosed by different criteria. Therefore, the composition of premenopausal and postmenopausal women might only have a minimal influence on the significance of this study to predict final pathology depending on preoperative MRI and grade assessment (endometrial biopsy) in low-risk EC patients.

## Conclusions

This ancillary study of a prospective, multicenter study found that a prediction of the postoperative pathology was ineffective in low-risk EC patients classified according to the presence or absence of MI and grade 1 or 2 on the preoperative assessment. Therefore, it is difficult to extend the current criteria for FST. On the other hand, the NPMs using Machine Learning provided a somewhat better prediction than conventional analysis. NPMs may help select more eligible patients for FST among low-risk EC patients. These NPMs and their outcomes should be confirmed in well-designed, large-scale prospective studies of premenopausal women.

## Supporting information

**S1 Fig. Comparison of the AUC performance based on the increased matching rate between preoperative and postoperative depth of MI.** Logistic Regression was used to predict Group 1. Reference represents conventional analysis. * denotes the matching rate of the depth of MI between preoperative assessment and postoperative pathology. ** indicates the matching rates that were randomly increased.
(PDF)

**S2 Fig. Comparisons of the imputed depth of MI and postoperative depth of MI.** a. NPM1, b. NPM2. Value 0 means no MI, value 1 means MI <1/2, and value 2 means MI ≥1/2. MI, myometrial invasion; NPM 1, New Prediction Model 1; NPM 2, New Prediction Model 2.
(PDF)

**S1 Table. Relationship between the preoperative assessment and postoperative pathology in a higher-risk population than a low-risk population based on the postoperative final pathology.**
(PDF)

**S2 Table. Raw data.**
(PDF)

## Author Contributions

**Conceptualization:** Dong-hoon Jang, Hyun-Gyu Lee, Banghyun Lee, Sokbom Kang, Jong-Hyeok Kim, Byoung-Gie Kim, Jae-Weon Kim, Moon-Hong Kim, Xiaojun Chen, Jae Hong No, Jong-Min Lee, Jae-Hoon Kim, Hidemich Watari, Seok Mo Kim, Sung Hoon Kim, Seok Ju Seong, Dae Hoon Jeong, Yun Hwan Kim.

**Data curation:** Banghyun Lee, Sokbom Kang, Jong-Hyeok Kim, Byoung-Gie Kim, Jae-Weon Kim, Moon-Hong Kim, Xiaojun Chen, Jae Hong No, Jong-Min Lee, Jae-Hoon Kim, Hidemich Watari, Seok Mo Kim, Sung Hoon Kim, Seok Ju Seong, Dae Hoon Jeong, Yun Hwan Kim.

**Formal analysis:** Dong-hoon Jang, Hyun-Gyu Lee, Banghyun Lee.

**Investigation:** Dong-hoon Jang, Hyun-Gyu Lee, Banghyun Lee.

**Methodology:** Dong-hoon Jang, Hyun-Gyu Lee, Banghyun Lee.

**Project administration:** Banghyun Lee.

**Resources:** Banghyun Lee.

**Software:** Dong-hoon Jang, Hyun-Gyu Lee.

**Visualization:** Dong-hoon Jang, Hyun-Gyu Lee.

**Writing – original draft:** Dong-hoon Jang, Hyun-Gyu Lee, Banghyun Lee.

**Writing – review & editing:** Sokbom Kang, Jong-Hyeok Kim, Byoung-Gie Kim, Jae-Weon Kim, Moon-Hong Kim, Xiaojun Chen, Jae Hong No, Jong-Min Lee, Jae-Hoon Kim, Hidemich Watari, Seok Mo Kim, Sung Hoon Kim, Seok Ju Seong, Dae Hoon Jeong, Yun Hwan Kim.

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
