## [Decision Letter · Decision Letter 0]

7 May 2024

PONE-D-23-43861Prediction of final pathology depending on preoperative myometrial invasion and grade

assessment in low-risk endometrial cancer patientsPLOS ONE

Dear Dr. Lee,

Thank you for submitting your manuscript to PLOS ONE. After careful consideration, we feel that it has merit but does not fully meet PLOS ONE’s publication criteria as it currently stands. Therefore, we invite you to submit a revised version of the manuscript that addresses the points raised during the review process.

**ACADEMIC EDITOR: Please respond to all comments point by point**

We look forward to receiving your revised manuscript.

Kind regards,

Ahmed Mohamed Maged, MD

Academic Editor

PLOS ONE

Additional Editor Comments:

An interesting well designed study. Some issues need clarifications

1. Add type of the study to the title

2. More details about inclusion and exclusion criteria are needed as women with other malignancies associated comorbidities ...........etc

3.Trial registration details

4. Sample size calculation need to be explained in more details

5. please add what the study adds to the existing knowledge and the future recommendations in details

Reviewers' comments:

Reviewer's Responses to Questions

**Comments to the Author**

1. Is the manuscript technically sound, and do the data support the conclusions?

Reviewer #1: Partly

Reviewer #2: Partly

Reviewer #3: Yes

2. Has the statistical analysis been performed appropriately and rigorously? 

Reviewer #1: Yes

Reviewer #2: Yes

Reviewer #3: Yes

3. Have the authors made all data underlying the findings in their manuscript fully available?

Reviewer #1: Yes

Reviewer #2: Yes

Reviewer #3: Yes

4. Is the manuscript presented in an intelligible fashion and written in standard English?

Reviewer #1: Yes

Reviewer #2: Yes

Reviewer #3: Yes

5. Review Comments to the Author

Reviewer #1: The main objective of this study is to exted the current criteria for FST in endometrial cancer, but the population in the study includes woman in post-menopause. I am afraid that the study population does not reflect the final population that the study is aimed at; can you better argue this aspect?

Reviewer #2: The manuscript is well written and interesting, addressing the important issue of identifying candidates with low-risk endometrial cancer and the possibility to preserve fertility.

As a radiologist, I find nothing specifically to comment on. The MRI methods are described in previous papers from the KGOG.

The present paper describes advanced mathematical methods on preoperative assessment for prediction of postoperative pathology to help extend the current criteria for fertility sparing therapy, which was found to be difficult.

The AUC was found to be highest for machine learning NPM2, however only moderately higher in Group 1, 2 and 4, and in fact lower in Group 3.

As the authors write, larger studies are needed to confirm the results.

Thus, I suggest a more humble Conclusion inserting the word "somewhat" and "may": On the other hand, the NPMs using Machine Learning provided a somewhat better prediction than conventional analysis. NPMs may help...

Reviewer #3: Thanks to today's technologies, prediction model studies such as those in this article are now very popular, and I think they provide very successful prediction (mostly).

Therefore, considering how useful these prediction models are in the field of health, I think this article should definitely be included in the journal.

Frankly, very good statistics have been made and it has been turned into a nice article.

6. PLOS authors have the option to publish the peer review history of their article (what does this mean?). If published, this will include your full peer review and any attached files.

Reviewer #1: No

Reviewer #2: **Yes: **Henrik Leonhardt

Reviewer #3: **Yes: **OMER DEMIR

---

## [Author Response · Author response to Decision Letter 0]

20 May 2024

Editor:

An interesting well designed study. Some issues need clarifications

1. Add type of the study to the title

The following words were added to the title: Prediction of final pathology depending on preoperative myometrial invasion and grade assessment in low-risk endometrial cancer patients: a Korean Gynecologic Oncology Group ancillary study.

2. More details about inclusion and exclusion criteria are needed as women with other malignancies associated comorbidities ...........etc

The inclusion criteria were already mentioned in detail as follows: The inclusion criteria were no MI or MI <1/2 on preoperative MRI and endometrioid adenocarcinoma and grades 1 or 2 on the endometrial biopsy.

This study did not have exclusion criteria. This study examined the relationship between the preoperative assessment and postoperative final pathology depending on the presence or absence of MI and grades 1 or 2 in patients with low-risk endometrial cancer (EC) on a preoperative assessment using the data from the KGOG 2015. Therefore, other factors, such as other malignancies associated with comorbidities, etc., did not have an influence on the results of this study. Therefore, exclusion criteria were not required.

Exclusion criteria of the KGOG 2015 were as follows: patients with histologic features suggesting squamous cell carcinoma or carcinosarcoma on preoperative biopsy; inadequate imaging study; no lymph node dissection; and sarcoma (reference 6 in this study).

Therefore, the following sentence was added: The inclusion criteria were... ; there were no exclusion criteria.

3. Trial registration details

The following words and contents were added: In KGOG 2015, between January 1, 2012, and December 31, 2014, 529 consecutive EC patients underwent a preoperative assessment based on MRI, an endometrial biopsy, and serum CA 125 testing, followed by surgical staging, including systemic pelvic and para-aortic lymphadenectomy [6]. In this prospective, multicenter cohort study, the participants were enrolled in 20 hospitals in three countries (Korea, Japan, and China) between January 2012 and December 2014. Approval from local institutional review boards was obtained for each center. Each participating center was a tertiary hospital that regularly provided surgical care for EC and had multidisciplinary teams that included specialized gynecologic oncologists, pathologists, and radiologists. The inclusion criteria were as follows: 1) EC, 2) no deep MI (MI <1/2) on MRI, 3) no enlarged lymph nodes on MRI, 4) no suspicious extrauterine spread, and 5) serum CA 125 < 35 U/mL. Patients with squamous cell carcinoma or carcinosarcoma on a preoperative biopsy, inadequate imaging study, no lymph node dissection, or sarcoma were excluded. The 2009 FIGO classification was used for the stage based on the final pathological findings.

4. Sample size calculation need to be explained in more details

In this study, the sample size was not calculated. All patients who were eligible for the inclusion criteria were included.

Processes to select total eligible patients and four individual groups were already provided in the manuscript in detail as follows: two hundred and fifty-one eligible patients were selected from the KGOG 2015 dataset according to the inclusion criteria, which were no MI or MI <1/2 on preoperative MRI and endometrioid adenocarcinoma and grades 1 or 2 on the endometrial biopsy. Then, the eligible patients for this study were assigned to four groups. Group 1 included patients with no MI on preoperative MRI and grade 1 on an endometrial biopsy. Group 2 included patients with no MI on preoperative MRI and grade 2 on an endometrial biopsy. Group 3 included patients with MI <1/2 on preoperative MRI and grade 1 on an endometrial biopsy. Group 4 included patients with MI <1/2 on preoperative MRI and grade 2 on an endometrial biopsy.

5. please add what the study adds to the existing knowledge and the future recommendations in details

This study already provided ‘what the study adds to the existing knowledge’ in details as follows: 1. a prediction of the postoperative pathology was ineffective in low-risk EC patients classified according to the presence or absence of MI and grade 1 or 2 on the preoperative assessment; 2. because the NPMs using Machine Learning have a somewhat better prediction than conventional analysis, NPMs may help select more eligible patients for FST among low-risk EC patients.

The following words were added to ‘the future recommendations’: These NPMs and their outcomes should be confirmed in well-designed, large-scale prospective studies of premenopausal women.

Reviewer #1: The main objective of this study is to exted the current criteria for FST in endometrial cancer, but the population in the study includes woman in post-menopause. I am afraid that the study population does not reflect the final population that the study is aimed at; can you better argue this aspect?

In this study, 39.4% of the population were premenopausal women, and 60.6% of the population were postmenopausal women. The average age of women diagnosed with endometrial cancer is 60. Endometrial cancer affects commonly postmenopausal women and between 15% and 25% of EC are found in premenopause [reference 14 in this study]. Therefore, it is difficult to investigate the study with only premenopausal women because of the low incidence of endometrial cancer in premenopausal women.

Recently, some studies have reported the difference between pre- and postmenopausal women with endometrial cancer. One study reported that women who were postmenopausal had worse tumor pathological gradings, more severe muscular invasion, and a higher rate of lymphatic metastasis. These factors led to a higher demand for postoperative radiotherapy in patients but a lower survival rate (reference: Lyu YL, et al., Transl Cancer Res. 2023 Mar 31;12(3):595–604). Another study reported that premenopausal women more often had ‘low’ risk disease (78% vs. 46%, p < 0.001). Among sonographic and anthropometric features, only an irregular endometrial-myometrial border was associated with ‘intermediate/high’ risk compared to ‘low risk’ in premenopausal women. Conversely, in postmenopausal women, multiple features are correlated with ‘intermediate/high’ risk disease. These findings emphasize the importance of considering menopausal status when evaluating sonographic features in women with endometrial cancer (Green RW, et al., Diagnostics (Basel), 2023, Dec 19;14(1):1).

To our knowledge, there is no evidence that in endometrial cancer, pathologic findings (endometrioid type or grade) or MRI findings differ between premenopausal women and postmenopausal women or are diagnosed by different criteria.

Therefore, we think that the composition of premenopausal and postmenopausal women might have only a minimal influence on the significance of this study to predict final pathology depending on preoperative myometrial invasion (MRI) and grade assessment (endometrial biopsy) in low-risk endometrial cancer patients.

However, our results need to be confirmed in the premenopausal population.

Therefore, the following sentences were added to limitations: Third, this study included premenopausal (39.4%) and postmenopausal (60.6%) women despite the objective of this study being to extend the current criteria for FST in EC. On the other hand, EC commonly affects postmenopausal women, and the incidence of EC in premenopausal women is low (15–25%) [14]. Therefore, it is difficult to conduct the study with only premenopausal women. Moreover, there is no evidence that in EC, the pathology findings (endometrioid type or grade) or MRI findings differ between premenopausal women and postmenopausal women or are diagnosed by different criteria. Therefore, the composition of premenopausal and postmenopausal women might only have a minimal influence on the significance of this study to predict final pathology depending on preoperative MRI and grade assessment (endometrial biopsy) in low-risk EC patients.

Reviewer #2: The manuscript is well written and interesting, addressing the important issue of identifying candidates with low-risk endometrial cancer and the possibility to preserve fertility. As a radiologist, I find nothing specifically to comment on. The MRI methods are described in previous papers from the KGOG. The present paper describes advanced mathematical methods on preoperative assessment for prediction of postoperative pathology to help extend the current criteria for fertility sparing therapy, which was found to be difficult. The AUC was found to be highest for machine learning NPM2, however only moderately higher in Group 1, 2 and 4, and in fact lower in Group 3. As the authors write, larger studies are needed to confirm the results. Thus, I suggest a more humble Conclusion inserting the word "somewhat" and "may": On the other hand, the NPMs using Machine Learning provided a somewhat better prediction than conventional analysis. NPMs may help... 

The following words were inserted according to reviewer’ suggestion: On the other hand, the NPMs using Machine Learning provided a somewhat better prediction than conventional analysis. NPMs may help select more eligible patients for FST among low-risk EC patients.

Memos in manuscript:

1. Introduction: ………….‘‘low-risk’’ population.

Please provide references to the literature to support this statement

References 2–5 were added. Therefore, in the entire manuscript, the number of references was changed.

2. Materials and Methods; 3.1.1. Data Preprocessing: First, missing values were identified in the preoperative tumor size (largest diameter).

Can you justify why did you have missing data about this parameter?

Unfortunately, we cannot clearly explain the reasons for the missing values occurrence in tumor size (largest diameter) on preoperative MRI because KGOG 2015 did not mention ‘preoperative tumor size (largest diameter)’ in its published article (reference 6 in this study).

This study was an ancillary study of KGOG 2015, which was a prospective, multicenter cohort study performed between January 1, 2012, and December 31, 2014. In KGOG 2015, all participating centers (20 hospitals in three countries) were tertiary hospitals that regularly provided surgical care for endometrial cancer and had multidisciplinary teams including specialized gynecologic oncologists, pathologists, and radiologists (reference 6 in this study). Because KGOG 2015 was a trial performed before 10 years, we cannot ask the radiologists the reasons. Therefore, we just guess that adequate MRI findings might have been provided by radiologists, but very small tumor sizes (< 1cm) might not have been reported.

We hope that you will understand this situation.

Can you quantify how many missing value imputed with mean post-operative value?

The number of missing values was already provided as the findings of the preoperative MRI in Table 1. All missing values were imputed with the mean post-operative value.

Therefore, the following words were added to the sentence: First, missing values were identified in the preoperative tumor size (largest diameter) (Table 1). All these missing values were imputed with mean postoperative tumor size values………………

---

## [Decision Letter · Decision Letter 1]

30 May 2024

Prediction of final pathology depending on preoperative myometrial invasion and grade

assessment in low-risk endometrial cancer patients: a Korean Gynecologic Oncology Group ancillary study

PONE-D-23-43861R1

Dear Dr. Lee,

We’re pleased to inform you that your manuscript has been judged scientifically suitable for publication and will be formally accepted for publication once it meets all outstanding technical requirements.

Kind regards,

Ahmed Mohamed Maged, MD

Academic Editor

PLOS ONE

Additional Editor Comments (optional):

Reviewers' comments:

Reviewer's Responses to Questions

**Comments to the Author**

1. If the authors have adequately addressed your comments raised in a previous round of review and you feel that this manuscript is now acceptable for publication, you may indicate that here to bypass the “Comments to the Author” section, enter your conflict of interest statement in the “Confidential to Editor” section, and submit your "Accept" recommendation.

Reviewer #2: All comments have been addressed

2. Is the manuscript technically sound, and do the data support the conclusions?

Reviewer #2: Yes

3. Has the statistical analysis been performed appropriately and rigorously? 

Reviewer #2: Yes

4. Have the authors made all data underlying the findings in their manuscript fully available?

Reviewer #2: Yes

5. Is the manuscript presented in an intelligible fashion and written in standard English?

Reviewer #2: Yes

6. Review Comments to the Author

Reviewer #2: (No Response)

7. PLOS authors have the option to publish the peer review history of their article (what does this mean?). If published, this will include your full peer review and any attached files.

Reviewer #2: **Yes: **Henrik Leonhardt

---

## [Editor Report · Acceptance letter]

18 Jun 2024

PONE-D-23-43861R1 

PLOS ONE

Dear Dr. Lee, 

I'm pleased to inform you that your manuscript has been deemed suitable for publication in PLOS ONE. Congratulations! Your manuscript is now being handed over to our production team.

Kind regards, 

on behalf of

Professor Ahmed Mohamed Maged 

Academic Editor

PLOS ONE